# Viral Infections and Systemic Lupus Erythematosus: New Players in an Old Story

**DOI:** 10.3390/v13020277

**Published:** 2021-02-11

**Authors:** Marco Quaglia, Guido Merlotti, Marco De Andrea, Cinzia Borgogna, Vincenzo Cantaluppi

**Affiliations:** 1Nephrology and Kidney Transplantation Unit, Department of Translational Medicine, University of Piemonte Orientale (UPO), “Maggiore della Carità” University Hospital, Via P. Solaroli 17, 28100 Novara, Italy; marco.quaglia@med.uniupo.it (M.Q.); guido.merlotti@maggioreosp.novara.it (G.M.); 2Center for Translational Research on Autoimmune and Allergic Disease-CAAD, 28100 Novara, Italy; marco.deandrea@unito.it (M.D.A.); cinzia.borgogna@med.uniupo.it (C.B.); 3Department of Public Health and Pediatric Sciences, Medical School, University of Turin, 10126 Turin, Italy; 4Department of Translational Medicine, University of Piemonte Orientale (UPO), 28100 Novara, Italy

**Keywords:** Epstein–Barr virus, parvovirus B19, retroviruses, human endogenous retroviruses, human immunodeficiency virus, transfusion-transmitted virus, cytomegalovirus, systemic lupus erythematosus, antiphospholipid syndrome, autoimmunity

## Abstract

A causal link between viral infections and autoimmunity has been studied for a long time and the role of some viruses in the induction or exacerbation of systemic lupus erythematosus (SLE) in genetically predisposed patients has been proved. The strength of the association between different viral agents and SLE is variable. Epstein–Barr virus (EBV), parvovirus B19 (B19V), and human endogenous retroviruses (HERVs) are involved in SLE pathogenesis, whereas other viruses such as Cytomegalovirus (CMV) probably play a less prominent role. However, the mechanisms of viral–host interactions and the impact of viruses on disease course have yet to be elucidated. In addition to classical mechanisms of viral-triggered autoimmunity, such as molecular mimicry and epitope spreading, there has been a growing appreciation of the role of direct activation of innate response by viral nucleic acids and epigenetic modulation of interferon-related immune response. The latter is especially important for HERVs, which may represent the molecular link between environmental triggers and critical immune genes. Virus-specific proteins modulating interaction with the host immune system have been characterized especially for Epstein–Barr virus and explain immune evasion, persistent infection and self-reactive B-cell “immortalization”. Knowledge has also been expanding on key viral proteins of B19-V and CMV and their possible association with specific phenotypes such as antiphospholipid syndrome. This progress may pave the way to new therapeutic perspectives, including the use of known or new antiviral drugs, postviral immune response modulation and innate immunity inhibition. We herein describe the state-of-the-art knowledge on the role of viral infections in SLE, with a focus on their mechanisms of action and potential therapeutic targets.

## 1. Introduction

Autoimmune disorders (AIDs) have a multifactorial pathogenesis, characterized by an interplay between immune dysregulation, environmental factors, and a genetic background [1].

Systemic lupus erythematosus (SLE) is a multisystem AID predominantly affecting women of child-bearing age, with a chronic relapsing–remitting course. Key aspects of SLE pathophysiology include impaired clearance of nucleic acids (NA), enhanced type I Interferon (IFN) response, abnormality in B cell tolerance and production of multiple autoantibodies, with immunocomplex formation and deposition causing progressive organ damage.

Antinuclear antigens (ANA), antidouble stranded DNA (dsDNA) and anti-Smith (anti-Sm) antibodies are serological hallmarks of SLE.

Clinical manifestations are variable. Constitutional, mucocutaneous and muscoloskeletal symptoms usually represent the earliest signs of disease. However, any apparatus can be involved, from the cardiovascular to the central nervous system, and an interval of years can exist between the onset of different symptoms [1,2].

Lupus nephritis (LN) develops in around 50% of patients and can progress to end-stage renal disease in 10% of them, significantly increasing disease morbidity and mortality [3].

Despite immunosuppressive therapy, currently based on hydroxychloroquine, mycophenolate mofetil, cyclophosphamide and steroids [1], one third of patients require multiple treatment cycles due to resistant or relapsing forms [4], with a heavy impact on all aspects of their lives [5]. The increasingly widespread use of biologics (Rituximab and Belimumab) is only partially overcoming these limits [1].

The establishment of a causal link between infections and autoimmunity has been largely evaluated in a host of clinical studies, proving the role of infectious agents in the induction, as well as in the progression or flareups of SLE. However, we are still far from an in depth understanding of microbial–host interactions in its pathogenesis.

An underlying trigger for SLE has remained elusive, and multiple interacting environmental and genetic factors likely contribute to onset and perpetuation. Among environmental influences, infectious agents appear to play a pivotal role in driving autoimmunity [6,7,8,9,10,11].

On the other hand, chronic immunosuppressive therapy significantly increases the risk of infections in SLE [12] and viral infections or reactivation can be severe and even life-threatening in this setting [13]. Thus, the relationship between viruses and SLE is complex and multifaceted.

The aim of this review is to provide state-of-the-art knowledge on the role of viral infections in triggering SLE onset and flares and in modulating its phenotype and course, with a focus on viral mechanisms of interaction with host immune system. The impact of immunosuppressive therapy on the risk of developing viral infections in SLE patients is beyond the scope of this work.

The PubMed library was searched from inception to December 2020, using a combination of Medical Subject Headings (MeSH) and keywords related to SLE, autoimmunity, AIDs and viruses, including: Epstein–Barr virus (EBV), Parvovirus B19 (B19V), Retroviruses (RVs), Human Endogenous Retroviruses (HERV), Human Immunodeficiency Virus (HIV), Torque Teno Virus (TTV), Cytomegalovirus (CMV). We also checked the references of relevant articles.

## 2. General Mechanisms of Virus-Induced Autoimmunity

Association of SLE with viral infection has been proposed on the basis of some key observations, which can be summarized as follows: (a) it can occur during chronic viral infections; (b) SLE onset of can be concomitant with that of a viral infection; (c) experimental evidence suggests that specific viral infections can trigger it. 

In general, viral infections can interact with the host immune system through several mechanisms which ultimately lead to the loss of tolerance, the production of autoantibodies, the tissue deposition of immune complexes and consequent tissue damage. These are: structural or functional molecular mimicry, epitope spreading, superantigen production, bystander activation, altered apoptosis and clearance deficit, epigenetic factors, persistent or recurrent viral infection, and innate immunity activation with type I IFN production [14,15,16,17,18,19,20,21,22,23,24,25,26].

All these mechanisms, described in Table 1, have been potentially implicated in the pathogenesis of SLE but are shared by many other AIDs. 

Molecular mimicry and epitope spreading are classical, well-documented mechanisms of autoimmunity induction, as confirmed in a recent study which showed massive peptide sharing between EBV epitopes and human proteins [15].

Over the last decade, however, epigenetic mechanisms have been highlighted as an important modality of the interaction between viruses and the immune system in SLE, as they allow viruses to create gene expression profiles predisposing the host to autoimmunity. They include DNA methylation changes in SLE-susceptibility genes (e.g., IFN-related genes), histone modifications and microRNA (miRNA) modulation [24,25,26]. These mechanisms are especially exploited by HERVs, as detailed in a specific paragraph.

Persistent viral infection can be due to viruses with the ability to evade immune clearance indefinitely, such as herpesviruses and hepatitis C virus (HCV). This determines chronic inflammation and polyclonal T- and B-cell proliferation, eventually leading to emergence of self-reactive clones and autoantibodies production. Many non-retrovirus RNA viruses cause “within host” persistent infections which can occasionally trigger AIDs. Specific strategies to achieve viral persistence have been partly elucidated for HCV, a virus associated with several autoimmune manifestations and possibly also with SLE, and will be analyzed in a specific paragraph (Section 3.1) [27,28].

Recurrent viral infections role in autoimmunity has been highlighted by recent evidence on type 1 diabetes (T1D) pathogenesis. Human pancreatic β-cells can sustain recurrent enteroviral infections, including RNA-virus Coxsackieviruses B (CVB), which can induce an adaptive response with paradoxical production of “facilitating antibodies” (antibody-dependent enhancement, or ADE). When CVB infection recurs, ADE effect facilitates monocyte recruitment, the type I IFN response and the dissemination of infection. Subsequent HERV-W envelope protein (HERV-W-Env) transactivation further promotes pancreatic autoimmune attack, leading to T1D [29]. Recurrent herpetic infections also appear to play a role in several postinfectious autoimmune neurological disorders. Herpes Simplex virus (HSV) infection can lead to Herpes Simplex Virus encephalitis, which evolves in autoimmune encephalitis in up to 27% of cases, while Varicella Zoster Virus (VZV) has been associated to Neuromyelitis Optica spectrum disorders [30]. 

Activation of innate immunity is another important mechanism facilitating autoimmunity (Table 1). Viral nucleic acids (NAs) and other pathogen- or damage-associated molecular patterns (PAMPs and DAMPs, respectively) interact with a variety of pattern recognition receptors (PRRs) such as toll-like receptors (TLRs), the nucleotide binding and oligomerization domain receptors (NLRs), retinoic acid-inducible gene-I (RIG-I) and melanoma differentiation associated gene 5 (MDA-5), known as RIG-I-like receptors (RLRs), and cyclic GMP-AMP synthase (cGAS). After binding viral NAs, these complexes can be incorporated into apoptotic blebs, promoting the activation of dendritic cells (DCs) and B cells, or activating multiple intracellular pathways converging on the expression of proinflammatory cytokines and type I IFN response genes in plasmacytoid DCs (pDCs) [31], as shown in Figure 1.

Thus TLRs, RLR and the cGAS receptor systems play an essential role in host defense against viral infections, but at the risk of potential autoreactivity against abundant self-NAs.

Of interest is the fact that the expression of TLR-7 and TLR-9 by B cells is associated with the production of typical SLE autoantibodies (anti-dsDNA, antiribonucleoproteins) [32,33].

Finally, direct cytotoxicity of viruses with a specific tropism for a certain tissue can result in development of an AID as a consequence of cell destruction. For example, herpesviruses are neurotropic and can kill central nervous system cells causing several AIDs [34] and, as already mentioned, CVB and HERV-W-Env exert a direct cytotoxic effect towards pancreatic β-cells, contributing to T1DM pathogenesis [29].

## 3. Role of Specific Viral Infections in SLE

Many viruses have been linked to the pathogenesis of SLE, particularly EBV, B19V, RVs and CMV. Among these, EBV has the strongest evidence as an etiological candidate, and a definite association seems to exist also with B19V and RV. The strength of association is variable for other viruses [14].

### 3.1. Epstein–Barr Virus (EBV)

The general features of EBV are outlined in Table 2. EBV is present in a latent form in the memory B cells of 95% of the world population. After being transmitted by saliva, it first infects pharyngeal epithelial cells and then resting B-lymphocytes, T lymphocytes, NK cells and neutrophils [35]. While a primary infection is usually asymptomatic (during childhood) or causes a self-limiting disease, infectious mononucleosis (IM) (in adolescence), the virus is also associated with several neoplasms and AIDs [6,7,8,9,10,11].

An in depth analysis of EBV’s molecular biology is beyond the scope of this work, but some of its features are relevant to understand the association between EBV and autoimmunity [55,56,57,58]. Briefly, EBV is characterized by a peculiar ability to shift between an active lytic cycle and a latent state from which it can reactivate, depending on its interaction with host immune cells [59]. A second key feature is that the virus can deeply alter physiological B cell differentiation to determine life-long infection in memory B cells [58,59,60].

These two phases are characterized by the expression of different viral proteins, some of which are involved in immune system evasion and autoimmunity induction (Table 3): (a)Lytic phase. It occurs during primary infection and subsequent EBV reactivations from the latent state. The viral DNA is replicated and most viral genes are expressed, allowing spread to other cells. Key lytic phase proteins are ZEBRA protein (BZLF-1), which can deregulate immune surveillance [61]; an IL-10-homologue virokine, which inhibits IFN-γ production, CD8+ cytotoxic T-cells and MHC-I expression [62]; restricted early antigen (EA/R), a viral homologue of Bcl-2 oncogene which makes infected epithelial and B-cells resistant to apoptosis [63].(b)Latent phase. It develops after primary infection, as a result of the host’s EBV-specific immune response. The expression of latent phase viral proteins is epigenetically controlled and results in an almost complete silencing of the EBV genome in memory B lymphocytes [64,65], thus avoiding recognition by T cells. Only a few genes are expressed [66], coding for three latent membrane proteins (LMPs)—LMP-1, LMP-2A and LMP-2B—and 5 EBV nuclear antigens (EBNA)—EBNA 1, -2, -3A, -3B and -3C. This limited set of proteins is critical for self-reactive memory B lymphocyte “immortalization”, autoimmunity induction [14] and EBV-driven oncogenesis [59].

LMP-1 and LMP-2 both help EBV gain access to the memory B cell pool and act as survival signals for EBV-infected lymphocytes, mimicking signals from CD40 transduction pathway and antigen-binding to B-cell receptor (BCR), respectively [66,67]. 

LMP2-A has been associated with the evasion of affinity-based selection in germinal centers [67].

LMP-1 has more complex functions, highly relevant to autoimmunity: it induces BAFF (B-cell activating factor of the tumor necrosis family), which rescues self-reactive B cells, it can determine a SLE-like form in transgenic mice [77] and has been associated with type I IFN pathway activation in SLE [68]. Of interest is the fact that there is a mutual interaction between LMP-1 expression and IFN-α levels. On the one hand, EBV-DNA and RNA can induce the production of IFN-α by pDCs through TLR 9- and 7-mediated stimulation of LMP-1 [69,70,71]; on the other hand, IFN-α and other proinflammatory cytokines can induce LMP-1 expression in EBV-infected B-cells. Recent evidence has also highlighted the role of LMP-1 in inducing Programmed Death-Ligand 1 (PD-L1) overexpression in EBV-infected cells such as neutrophils [35]. Taken together, these data highlight the involvement of LMP-1 in key pathways modulating SLE activity: IFN-α, TLRs and PD-1/PD-1L system (Figure 1).

An increased expression of LMP-1 and EBV-encoded RNA-1 (EBER-1) has also been found in the renal tissue of patients with LN as compared to HCs and minimal change nephropathy patients [72] and a correlation between the degree of LMP-1 expression and the histological severity of LN has been suggested [73].

EBNA proteins also play a key role in EBV latent phase and autoimmunity induction. 

EBNA-1 is the only viral antigen which acts as a replication factor during latent infection and it can be the only EBV protein expressed in resting B cells. It is not degraded by proteasome and consequently cannot be presented at the cell surface, becoming invisible to immune system. EBNA-1 cross-reacts with several self-antigens and can induce autoimmunity through molecular mimicry with C1q and epitope spreading [74,75].

EBNA-2 controls the expression of all other latent viral genes and blocks replication in most EBV-infected cells. Genomewide association studies (GWAS) have shown a high number of associations (>2.200) between hundreds of transcription factors (TFs) and many of the genetic loci increasing the risk of SLE. Of interest is the fact that around 50% of these loci can be occupied by EBNA-2, suggesting a dominant role of this protein in modulating susceptibility not only to SLE but also to other AIDs, such as rheumatoid arthritis (RA) and multiple sclerosis (MS), which could be classified as “EBNA-2 disorders” [76].

### Alterations of EBV-Specific Immune Response in SLE

Peculiar alterations in EBV-specific immune response are found in SLE (Table 4).

Both lytic and latent EBV proteins elicit a strong T- and B-cell response. The expansion of EBV-infected B cells is physiologically controlled by CD8+ cytotoxic T lymphocytes (CTL). EBV-specific CD8+ CTL are instead dysfunctional in SLE and exhibit an abnormally reduced production of cytokines when stimulated with EBV [78,79,80]. This inadequate response favors recurrent lytic EBV replication, resulting in a chronically increased blood viral load and enhanced cell death [75,81,82]. T-cell exhaustion due to increased expression of PD-1 and PD-1L is probably an important mechanism through which EBV persistent infection can blunt CTL response and evade immune control [83]. A decreased Th17 and Treg response may be another key EBV-induced alteration [84].

Persistently active viral cycles increase the risk for both EBV-associated malignant transformation and AID in genetically predisposed individuals. An inverse correlation exists between impaired function of EBV-specific CTL, serum levels of antibodies against lytic antigen (anti-EA/D) and disease activity (SLEDAI)/autoantibodies [78,79,80]. This reflects an unbalanced B/T cell-mediated response towards EBV and other viruses, predisposing the host to an enhanced frequency of EBV reactivations and autoimmunity induction.

The impaired control of EBV by CTL and persistent latent EBV infection constitute the primum movens of a broad spectrum of immunological alterations which characterize SLE (Table 4).

The most immediate one is that elevated EBV viral loads are found in peripheral B cells [85] and in PBMC of SLE patients, with a 10-to 40-fold increase compared to HCs, regardless of immunosuppressive therapy [82,86,87].

Furthermore, the aberrant expression of key viral mRNAs related to proteins in both the lytic (BZLF-1, BLLF-1) and latent phases (LMP-1, LMP-2 and EBNA-1 in PBMCs) suggests intense EBV reactivation and enhanced survival of EBV-infected lymphocytes [85,88]. 

Persistently active EBV infection due to inefficient T cell response appears to be associated with a shift towards an amplified humoral response against EA/D and other viral antigens [63,80,89]. Several features of this abnormal humoral response to EBV are typical of SLE (Table 4).

First of all, the seroprevalence of anti-EBV antibodies is very high in SLE [90,91,92]. Two recent metanalysis have confirmed a strong association between EBV seropositivity (EBNA-1 and VCA IgG) and SLE onset [93,94], indicating that prior EBV infection is a prerequisite for developing SLE, at least in some ethnic groups.

Second, the humoral response is qualitatively different from that of HCs in several respects. These peculiarities were demonstrated in the clinical studies summarized in Table 4.

Among these, it is interesting to note that the prevalence of an IgA response against EBV antigens is elevated in SLE patients compared to HCs and also other AIDs [99,100].

Furthermore, the coexistence of multiple antibody isotypes against EA/D is significantly more frequent in SLE. In a study by Draborg, 65% of SLEs tested positive for two or more isotypes of antibodies against EA/D, whereas none of the HCs were positive for three isotypes, only 10% were positive for two isotypes and 65% of them had no antibodies against EA at all [89]. These alterations likely reflect the chronic production of antibodies against cells expressing EA/D, EBV-infected cell lysis and the release of EA/D and intracellular antigens. Depending on the type of infected cell, this could result in the enhanced production of either IgA (epithelial cells) or IgG (lymphocytes) against EA/D, in addition to autoantibodies (EA/D is bound to dsDNA, functioning as EBV-DNA polymerase). Thus, the presence of multiple anti-EA/D antibody isotypes could reflect a more disseminated infection, with a higher risk of cell lysis and autoimmunity induction.

In conclusion, the role of EBV in the pathogenetic mechanisms of SLE is supported by very high seroprevalence of antibodies against EBV, increased viral loads and peculiar alterations of the EBV-specific immune response. Overall, these data indicate that previous exposure to EBV represents a triggering factor for autoantibody generation and clinical disease onset in genetically predisposed individuals. 

The presence of EBV proteins in renal tissue from patients with LN confirms direct involvement of this virus in organ damage. 

### 3.2. Parvovirus B19 (B19V)

B19V is a small single-strand DNA virus with a worldwide distribution. Its general features are outlined in Table 2. Although it usually has a benign, self-limiting course, B19V proteins can induce SLE and other AIDs [106,107].

Most viral proteins are involved in inflammatory and autoimmune disorders: VP-1, which includes a “VP-1 unique region” (VP-1u) characterized by a phospholipase A2-like activity, probably playing a role in viral entry into the cell [108] and the generation of inflammatory mediators (leukotrienes and prostaglandins) and phospholipids epitopes, potentially triggering antiphospholipid (aPL) antibodies [109]. Of note is the fact that VP1-unique region contains many epitopes recognized by neutralizing antibodies, which determine life-long protection [106].NS-1, a cytotoxic, nonstructural protein which can transactivate several promoters, also enhancing retroviral replication, induces IL6 gene expression and stimulates polyclonal B-cell activation [110]; the expression of NS-1 probably marks complicated or chronic B19V infections [111].

B19V has been associated with immune-mediated manifestations such as different types of glomerulonephritis (GN) (mainly acute postinfectious, membranoproliferative, SLE-like GN and minimal change disease) [112,113]. However, B19V seroprevalence among patients who undergo kidney biopsy is similar to the overall population and the mechanism of renal damage is still undefined, although it is likely mediated by endothelial injury [113].

As for systemic autoimmunity, acute B19V infection can transiently mimic full-blown SLE (facial rash, photosensitivity, acute polyarthritis, cytopenia) [114] or even trigger or exacerbate full SLE [115,116] and can be accompanied by a broad spectrum of autoantibodies, including antinuclear antigen antibodies (ANA), anti-dsDNA, rheumatoid factor and APL antibodies [115,117,118]. 

The abnormal apoptosis of B19V infected bone marrow (BM) erythroblast, with consequent production of many altered self-antigens, has been proposed as a possible mechanism favoring these autoimmune processes [119]

A specific aspect is B19V’s role in antiphospholipid syndrome (APS) pathogenesis. APS refers to the association of thrombosis and/or pregnancy morbidity with antiphospholipid (aPL) antibodies and can be either primary or develop within the setting of SLE (secondary APS) [120]. 

Although the transient appearance of APL antibodies is in general quite common during infections, the association between B19V, isolated APL antibodies (Lupus Anticoagulant or LAC, anti-β 2-glycoprotein I, anticardiolipin) and even full-blown APS has been consistently reported over the last two decades [121,122]. 

In a recent study by Hod T. et al. a correlation between viral serology and clinical and serological parameters was ruled out in the general SLE population, but a higher B19V seroprevalence (both IgG and IgM) was found in a subset of SLE patients with APS. These patients also had a higher titer of IgG antibodies to B19V compared to those without APS, suggesting a link between previous viral exposure and APS [42]. The role of protein VP1u as an antigen triggering APS-like syndromes has been established [109].

Recent studies also indicate that B19V is associated with a specific SLE complication, dilated cardiomyopathy (DCM), probably through an increase of Th-17-related cytokines, as prevalence of anti-B19-NS1 and anti-B19-VP1u IgG was significantly higher in patients with DCM than in the general SLE population [123].

In conclusion, although no study has formally proved a direct causal role of B19V in AIDs so far, as available evidence is largely based on small case series [109], there appears to be sufficient epidemiological evidence to conclude that B19V infection can trigger SLE and modulate its clinical expression [124]. The pathogenetic mechanisms of B19V infection (the promotion of erythroblast apoptosis and the expression of viral proteins VP-1 and NS-1, both characterized by important immunomodulatory properties) support its role as an inducer of a broad range of autoimmune phenomena, ranging from isolated autoantibodies to definite SLE and APS. 

### 3.3. Retroviruses

Both endogenous and exogenous RVs (Table 2) can stimulate the immune system and have been associated with SLE. Three elements indicating a relationship between RVs and SLE have long been recognized [125,126]:(a)The presence of retroviral-like particles in the tissues of SLE patients;(b)The presence of antiretroviral antibodies in an important proportion of SLE patients;(c)Clinically overlapping features between retroviral infections and SLE.

However, the relationship between RVs and SLE is multifaceted and must be differentiated according to the type of RV. While the association with HTLV-1 is still undefined and debated, being largely based on small studies or case-reports, the one between HIV and SLE is definite and plausibly supported by clinical and translational elements. 

As for HERV, a large body of evidence points to an important role in mechanisms of autoimmunity, ranging from molecular mimicry to the activation of intracellular sensors by viral RNA and epigenetic modulation of immune response genes. 

We will therefore separately analyze the different types of RVs. 

#### 3.3.1. Human Endogenous RVs (HERVs)

Up to 200 families of HERVs have been described and some of them (HRES-1, ERV-3, HERV-E 4-1, HERV-K10 and HERV-K18) have been especially implicated in SLE [54,127,128,129,130].

HERVs are ancient RVs which are stably integrated in host mammalian genome. Endogenous retroviral sequences (ERS) account for up to 8% of it and probably represent the remnants of exogenous RVs which integrated into the genome and became trapped during evolution, due to mutations of essential genes [54,131]. Environmental factors (such as infections, estrogens, drugs/chemicals, UVB exposure) can epigenetically reactivate the transcription of HERV-related sequences. Hence, HERV may represent the missing molecular link between the genetic predisposition to autoimmunity and external triggers [25]. Epigenetic changes are especially important as transducers of external stimuli: the genome is overall undermethylated in SLE, causing increased transcription of IFN-responsive genes and related ERSs [132].

While exogenous RVs are infectious and require a replication cycle to integrate proviral DNA into the host genome, endogenous RVs are genetically transmitted through the germline as proviral DNA. Despite some beneficial effects (e.g., the epigenetic modulation of immune response genes to fight exogenous viruses), HERV elevated transcription and anti-HERV antibody reactivity are strongly implicated in SLE pathogenesis [54,125,126]. GWAS has identified 124 unique ERV loci that are significantly more expressed in the PBMCs of SLE patients [133]. For example, HTLV-1-related ERS (HRES-1) is integrated at chromosome 1q42, a SLE-susceptibility locus, and codes for HRES/Rab4, a GTPase markedly overexpressed in T cells of SLE patients [134]. 

HERVs can trigger autoimmunity through several mechanisms [125,126]: (a)They produce neosynthesized viral antigens, which can stimulate autoimmune response through the molecular mimicry of self-proteins, determining the production of polyclonal antibodies and nephritogenic immunocomplexes; in some cases, they act as superantigens.(b)They can directly stimulate intracellular sensor molecules (e.g., TLRs) through their NAs, triggering inflammatory cascades.(c)They promote the transcription of IFN-γ-related and other immune-related genes. This epigenetic control is often mediated by HERV retroviral long terminal repeats (LTRs) domains, which are co-opted by their mammalian hosts as specific regulatory sequences.

Over the last decade, significant evidence has been accumulating on each of these mechanisms, which we will analyze in more detail.

(a)Molecular mimicry and production of pathogenic antibodies.

HERV proteins are characterized by strong cross-reactivity to self-antigens and molecular mimicry is well documented [99]. For example, HTLV-1-related endogenous sequence (HRES-1), can code for HRES-1/p28, a 28-kDa nuclear autoantigen targeted by cross-reactive antiviral antibodies [135].

This protein is characterized by a sequence which harbors a triplet of highly charged amino acids, which is also found in the gag-like region of 70K U1snRNP SLE autoantigen. These cross-reactive epitopes can drive the formation of antinuclear autoantibodies [129]. Furthermore, HRES-1/Rab 4 overexpression interferes with endosomal recycling of CD3/TCRζ chain in lupus T cells, determining a critical loss of expression of this receptor, and lowering T-cells’ threshold for activation (“lupus T cells”) [136,137]. 

In addition to exerting molecular mimicry, viral antigens can also trigger the production of antibodies and immune complexes, directly contributing to organ damage. Anti-HERV-K envelope antibodies have been identified in SLE and the envelope glycoprotein 70 (gp70) and viral NA-protein complexes are targeted by antibodies as a result of TLR-7 activation. Retroviral gp70-anti-gp70 immune complexes are nephritogenic in murine models of LN [138]. The synergistic effect of three different gp70 loci (Sgp3,4 and 5) enhances the expression of HERVs and of this protein [139] and germline deletions of SLE. Susceptibility Locus Sgp3, which encodes the HERV suppressors named “SNERV” 1 and 2, indirectly determines autoantigen gp70 overproduction and LN in SLE-prone mice [140]. 

Furthermore, the antibody-induced extensive internalization of retroviral envelope glycoproteins initiates signaling cascades leading to the activation of lymphocytes in a murine model [141].

(b) Direct stimulation of intracellular sensor molecules by viral NAs. 

Several NA-sensing systems are highly expressed in SLE PBMCs and can be activated by RVs RNA. Upon binding with viral NA they trigger a variety of pathways, all converging on type I IFN response [32].

RVs can be detected by several systems such as endosomal TLRs, which can recognize viral dsRNA (TLR-3) and ssRNA (TLR-7 and -8), in addition to dsDNA (TLR-9) [142]. 

The NAs of RVs can also sensitize other PRRs in the cytosol besides TLRs in the endosomes [143] (Figure 2).

While NA-sensing system is physiologically required to control viral infections, an excessive activation can drive SLE progression by enhancing the production of key cytokines, such as IL17A [144]. For example, a deficit in apoptosis can favor cell engulfment with HERV-derived DNA and abnormally amplify this process [125,145]. 

(c) Epigenetic control of host immune genes and other mechanisms of immunomodulation. 

HERV sequences coding for viral genes *gag, pol* and *env* are flanked by long terminal repeats (LTRs). 

LTRs are often present as solitary sequences, spread in high numbers of copies throughout the human genome and likely control the transcription of adjacent genes. A part of them have preserved regulatory activity, as they contain promoter/enhancer sequences which generally have a minor impact on gene transcription. However, alterations in the epigenetic mechanisms which normally control them can pathologically increase their activity. This epigenetic reset can occur in autoimmunity and cancer and promote the transcription of LTR-dependent immune genes and the reactivation of HERV. Exogenous viral infections (HIV, HTLV-1, influenza A virus, herpesviruses) can also rescue HERVs from their inactive status [125].

For example, MER 41, an LTR belonging to a family of HERVs, is a unique STAT1-binding site near the IFN-γ-inducible *Absent in Melanoma 2 (AIM2)* gene and acts as its promoter. AIM2 codes for a protein which is a key foreign DNA sensor, connected to innate response (AIM2 inflammasome) and SLE pathogenesis (Figure 2). Thus, uncontrolled retroviral regulatory sequences can represent a reservoir of IFN-inducible enhancers [131]. 

The role of different LTRs in modulating expression of both HERV genes and their molecular targets is further underlined in another setting. Increased HERV-E clone 4-1 mRNA expression was found in SLE CD4+ T-cells and PBMCs, secondary to a variety of factors: activation of Ca^++^/calcineurin (CaN)/Nuclear factor of activated T cells 1 (NFAT1), due to UVB or foreign infections with high IL6 and TNFα levels; Estrogen receptor-α (ER-α) signaling pathway; abnormal DNA hypomethylation of HERV-E clone 4-1 5’LTR. Once expressed, this HERV clone 3’LTR restrains miR-302d activity (an inhibitory miRNA, abnormally downregulated in SLE monocytes), activating a cascade which leads to DNA hypomethylation, IL17 release from CD4+ T-cells and type I IFN response [146,147,148]. Increased expression of HERV-E clone 4-1 *gag* transcripts has been associated with immunological activity and anti-U1RNP and anti-Sm autoantibodies [149]. This cascade is outlined in Figure 2. 

Overexpression and LTR polymorphisms of another viral protein, HRES-1/Rab4, which regulates the surface expression of CD4 on T cells through increased lysosomal degradation of CD4 and CD3/TCRζ, have been associated with T cell activation upon TCR stimulation. Similar to HERV-E clone 4-1, also HRES-1/Rab4 expression is induced by UVB, Estrogen and even by TAT protein of HIV-1, suggesting interactions between different RVs [129,134].

Another modality of the epigenetic modulation of immune gene expression occurs through Long Interspersed Nuclear Elements (LINEs), retrotransposable elements coding for endonuclease and retrotranscriptase. Due to their autonomous capacity to replicate and shift along the genome they can control in cis expression of adjacent key genes (innate immunity, apoptosis), in a similar way to intragenic LTRs [125]. Furthermore, their DNA and RNA transcripts can activate TLR-7 and -9, initiating pathways which converge on IFN gene transcription, especially in LN [150]. LINEs sequences are also derepressed by hypomethylation, as shown in SLE neutrophils and PBMC, and this process correlates with immunological activity [151,152]. 

In addition to epigenetic control, HERV can exert immunomodulation through proteins with superantigen properties. For example, the env protein of HERV-W (syncytin) and HERV-K18.1 may act as superantigens and strongly stimulate T lymphocyte proliferation [153]. 

However, HERVs may also play an immunosuppressive role in SLE. A HERV locus coding for envelope protein 59 negatively correlates with interleukin-6 (IL6) and TLR-7 expression. This protein contained an immunosuppressive domain (ISU) which showed strong anti-inflammatory properties in animal models and also in ex vivo experiments in SLE patients, suggesting a potential therapeutic application [154,155,156].

#### 3.3.2. Exogenous RVs: Human T-Cell Lymphotropic Virus Type I (HTLV-1) and Human Immunodeficiency Virus (HIV) 

Exogenous retroviruses, like HIV and HTLV-1, have been both associated with SLE-like manifestations or to a fair SLE diagnosis. Molecular mimicry has long been proposed as a pathogenetic mechanism, as cross-reactivity of HTLV-1 and HIV gag antigens with self-proteins exists.

##### HTLV-1

HTLV-1 can determine adult T cell leukemia/lymphomas and HTLV-1–associated myelopathy/tropical spastic paraparesis (HAM/TSP), which has been associated with several AIDs including SLE [51,157].

Anti-HTLV-1 antibodies have been detected in variable proportions of SLE patients and a single study suggested that seropositive patients could have a later-onset, milder disease [158].

HTLV-1 has the potential to induce SLE through molecular mimicry between a 28Kda protein coded by HRES-1, which acts as a nuclear autoantigen, and HTLV-1 gag p24 protein [159].

In addition, HTLV-1 codes for several regulatory proteins which modulate inflammatory pathways upregulating T CD4+ Th_1_ lymphocytes and downregulating Treg [51].

##### HIV

Epidemiological data suggest an association between HIV and SLE. In a large nationwide French study, a wide spectrum of autoimmune manifestations has been reported to occur during HIV infection, with an overall prevalence of 4.13%; they mainly included arthralgias or arthritis, psoriasis, sarcoidosis, RA and ankylosing spondylitis, while the prevalence of SLE was lower. These disorders mainly occurred in patients on antiretroviral therapy with undetectable HIV viremia, on average more than 10 years after HIV diagnosis [160]. Full-blown SLE appearing in HIV-positive patients, mostly Black Africans, have also been reported in smaller studies [161,162]. 

Furthermore, “LN-like” proliferative GN has been described as part of the spectrum of HIV-associated nephropathy (HIVAN). The availability of HAART may have contributed to the moderate prevalence of these manifestations over the last few decades and to remission induction of SLE-like GN in some cases [154,163]. 

Several other elements also support the plausibility of an association between HIV and SLE.

These two conditions share common clinical and laboratory manifestations and are both characterized by an activation of type-I IFN-mediated response, which is essential to counteract HIV infection but also represents a molecular “signature” of SLE. In HIV polyclonal proliferation of B cells is associated with hypergammaglobulinemia and production of broadly neutralizing antibodies against Env epitopes. Some of them, such as CH98, cross-react with ds-DNA. Interestingly, the subset of anti-dsDNA-positive HIV patients appears to be less prone to AIDS progression, suggesting that an amplified humoral response may be protective, at the price of an increased risk of inducing autoantibody production [164]. Anti-Sm and, more commonly, ANA and anti-aPL antibodies are also found in HIV [165]. 

Another shared key alteration is an impaired autophagy, with consequent accumulation of senescent or damaged cell components. Autophagy is a crucial mechanism to control HIV infection, through stimulation of TLR-7 and TRL-9 and activation of intracellular inflammatory pathways; however, HIV proteins can blunt this process to prevent clearance of virions, thus creating a predisposing condition for autoantigen exposal and SLE induction [166,167]. Of interest is the fact that the mammalian Target Of Rapamycin (mTOR) inhibitor rapamycin induces autophagy, with beneficial effects on both conditions [168]. 

### 3.4. Torque Teno Virus (TTV)

TTV is single-stranded circular DNA Anellovirus, originally named after the initials of the first patient (TT), then “transfusion-transmitted” (it was initially associated with posttransfusion hepatitis) and, more recently, “Torque Teno” virus (Table 2). It is considered a commensal human virus, ubiquitously infecting human populations by early infancy and has not been causally associated with any disease until recently. 

However, increased TTV load is common in AIDS patients and immunosuppressed solid organ transplant (SOT) recipients and this virus is also probably involved in SLE pathogenesis, even though Anelloviruses may represent a generic marker of immune dysfunction and immunodeficiency [44,169,170]. 

TTV-DNA was actually detected more frequently in SLE patients than in HCs and RA [129]. TTV PCR-positive SLE patients have a significant prevalence (21%–52%) of antibodies against HRES-1/p28 and molecular mimicry between this antigen and 70KU1snRNP may contribute to the generation of nuclear autoantibodies [159]. In addition, HRES-1/p28 cross-reacts with TTV peptide ORF2a [43,44,135] and also with EBV antigens (EBV-LF3 and EBNA 3 C) [171,172].

Finally, TTV-DNA could also activate innate immunity through TLR-9 [169]. 

### 3.5. Cytomegalovirus (CMV)

Human CMV is a member of the β-subgroup of the herpesvirus family, commonly acquired during childhood, with absent or self-limiting symptoms (Table 2). It is characterized by several features which can explain its involvement in autoimmunity and are similar to those of EBV: lytic replication in multiple tissues, lifelong persistence through periods of latency and intermitting reactivation, extensive manipulation of adaptive and innate immunity [173,174,175] (Figure 1).

A causal relationship between CMV and SLE and other AIDs is suggested by the literature. However, a clear association of CMV seroprevalence and disease has not been firmly established yet. Growing evidence suggests that CMV can have multiple interactions with the host immune system and could play a contributive role in flareups and possibly in conferring SLE specific-clinical features, such as associated secondary APS and Raynaud’s phenomenon. Population-based, long-term prospective studies are needed to better elucidate its impact on disease course [174].

Some key viral proteins mediate immunopathogenic mechanisms of CMV-induced autoimmunity:Phosphoprotein 65 (pp65_422–439_). This protein contains an epitope region which has a homology with TATA-box binding protein associated factor 9 (TAF9_134–144_). Consequently, antibodies against pp65_422–439_ cross-react with TAF9_134–144_ and also with ANA and anti-dsDNA in immunized BALB/c mice, in which they determine typical histological LN lesions. Furthermore, SLE patients appear to have higher anti-TAF9 antibody titers compared to HCs. Overall pp65 acts can induce autoantibodies production in susceptible animals [176,177].UL44 is nonstructural intranuclear protein which is essential for CMV replication. A human monoclonal antibody from CMV seropositive donor immunoprecipitates UL44 with other SLE nuclear antigens including nucleolin, dsDNA and ku70. UL44 appears to be relocated at cell surface complexed with these molecules during CMV-induced apoptosis, suggesting a mechanism of bystander autoantigen presentation [178].US31 is highly expressed in PBMC of SLE patients and can skew macrophage differentiation toward an M_1_ phenotype, activating inflammation through a direct interaction with NF-κB2 [179].

Potential cross-reactivity of CMV-specific T cells with La (SS-B) protein, a ribonucleoprotein associated with RNA polymerase, has been shown in childhood onset SLE [104,180].

Furthermore, CMV has the potential to extensively manipulate host T cell function during active infection. In vitro studies showed that CMV antigens can increase CD4+/CD8+ T-cell ratio in SLE PBMCs and their IL-4 and IFN-γ production, driving the development of a large pool of memory T-cells which can facilitate autoimmunity [181].

However, other studies showed that the dysfunctional EBV-specific CD4+ T-cell response observed in SLE patients does not seem to apply to CMV to the same extent [182] and that CMV-specific CD8+ T-cell responses appear preserved [183]. Overall, these data suggest that CMV probably plays a less important role in the development or exacerbation of SLE than EBV. Although the SLE-intrinsic dysregulation in B- and T-lymphocytes’ responses is the main reason for skewed EBV and CMV-specific T response [137], the intensity of immunosuppressive therapy also crucially modulates the risk of herpesviruses reactivation [184].

A peculiar aspect of CMV is that it can drive expansion of a highly cytotoxic, immunosenescence-associated subset of CD4+ T-cells lacking CD28 (CD28^null^CD4+). This T cell subset is resistant to CD4+CD25+T reg-mediated inhibition and to apoptosis and is probably mainly CMV-specific. While its frequency is usually below 1% even in very old people, CMV infection can increase it more than 10-fold. The accumulation of CMV-specific CD28^null^ infiltrates was demonstrated in disease affected tissues in SLE and other AIDs and has also been associated with atherosclerotic vascular complications in RA [185]. 

Finally, due to its tropism for endothelial cells, CMV has been implicated in the pathogenesis of vascular damage in systemic sclerosis (SSc), through molecular mimicry of normally expressed endothelial cell surface molecules [186], Raynaud’s phenomenon [187] and APS [188,189]. 

### 3.6. Other Viruses

We will separately analyze other RNA- and DNA-viruses potentially associated with SLE.

#### 3.6.1. HCV, Measles, Influenza A and Dengue Virus

HCV is a hepatotropic RNA-virus supposed to have an additional extrahepatic tropism, including that for B lymphocytes. This virus determines a chronic infection in 70% of patients, evading innate and adaptive immunity and promoting B lymphocyte proliferation and a low-grade, chronic systemic inflammation. HCV strategies to achieve viral persistence include: hijacking cellular factors such as miRNA that bind its genome protecting it from degradation, inhibiting NA detection by innate immunity sensors, limiting type I IFN response activation and repressing viral transcription and replication, thus allowing host cell survival [28,190]. 

Chronic HCV infection has been associated with a large spectrum of extrahepatic autoimmune manifestations, including cryoglobulinemic vasculitis, Sjögren’s syndrome, RA and SLE. Direct-acting antiviral agents and new B-cell-depleting or B-cell-modulating monoclonal antibodies will probably expand treatment options for HCV-related AIDs [191].

In a recent study on patients with SLE, RA and systemic sclerosis, IgG-seropositivity for measles virus (MeV) was associated with significantly higher titers of natural autoantibodies against citrate synthase, which have a modulating effect on the immune system [192].

A recent Korean study showed a significant association between seasonal influenza infection and flares in SLE patients [193]. A massive peptide overlap between five common human viruses (Influenza A virus, Borna disease virus, MeV, mumps virus and rubella virus) and human proteome emerged. Sequence similarity may explain autoimmune cross-reactions during immune responses directed towards viral infections or due to immunizations [194]. Furthermore, Influenza A virus can also rescue HERVs from their inactive status [125]. 

The relationship between Dengue virus (DENV) and SLE is complex and still remains to be elucidated. On the one hand, DENV has been anecdotally reported to trigger SLE and LN [46]; on the other hand, IgG that can cross-neutralize DENV are often found in SLE sera, probably explaining rarity of severe forms in SLE patients [195].

#### 3.6.2. Human Papillomavirus 

Human Papillomavirus (HPV) is a dsDNA virus which has recently been proposed to play a role in SLE pathogenesis on the basis of several elements: HPV seroprevalence is increased among SLE patients, SLE development post-HPV vaccination has been reported and a large peptide overlap has emerged between HPV L1 proteins and many human proteins such as lupus Ku autoantigen proteins p86 and p70, lupus brain antigen 1 homolog and others. This further supports the hypothesis of cross-reactivity as primum movens for SLE onset and the concept of SLE as “an autoimmune mosaic syndrome” [196].

## 4. Future Perspectives

The expanding knowledge of the role of viral infections or reactivation as a trigger for AID onset or as modifiers of disease severity and course is paving the way for alternative approaches in the management of SLE. As viral infections and autoimmunity share activation of overlapping pathogenetic pathways, there is a rationale in investigating the use of antiviral drugs as part of SLE therapy, along with modulation of virally induced immune response. Progress in these research areas will be analyzed. Although most data are preliminary and deserve to be confirmed, they represent frontiers which may lead to new therapeutic, diagnostic, and prognostic tools. 

### 4.1. Anti-EBV Therapies

An interesting potential therapeutic strategy could be that of modulating anti-EBV response. This approach has been investigated mainly in the setting of EBV-associated cancer, with the rationale of exploiting EBV presence in tumor cells to target them selectively [197,198]; however, it could also open new perspectives on AIDs.

Inhibitors of EBV reactivation and replication include drugs already licensed for the treatment of other herpesviruses and nutritional constituents [199]. 

The adoptive transfer of EBV-specific transgenic T cells has been therapeutically explored with clinical success in EBV-associated malignancies [200]. 

### 4.2. Anti-B19V Therapies

New antiviral strategies have been explored in recent years, including: hydroxyurea; cidofovir and brincidofovir, nucleotide analogues both characterized by a broad-range activity against DNA-viruses; coumarin derivatives and flavonoid molecules [201].

B19V nuclear localization signals (NLSs), which are critical to guide the virus into the nucleus, are promising therapeutic targets. Therapeutic vaccines or antiviral drugs interfering with them could block infection [202].

### 4.3. Antiretroviral Therapies

Reverse transcriptase inhibitors represent the cornerstone of antiretroviral agents employed in HIV therapy. A large study on over 20,000 patients from Taiwan has shown that HAART therapy could protect from SLE onset [203] and blunt reactivation of some HERVs [204], thus interfering with an important mechanism of epigenetic modulation in SLE. 

### 4.4. T Lymphocyte Modulating Therapies

Therapies targeting CD28^null^ T cells, a subset strongly associated with CMV infection, appear promising as they seem to represent a common pathogenic mechanism shared by SLE and other AIDs [185]. For example, anti-TNFα and methotrexate therapy reduce expansion of CD28^null^ CD4+ T-cells in RA and lipid-lowering rosuvastatin induces their apoptosis in patients with coronary heart disease. Antiviral ganciclovir has also proved effective in this regard in ANCA-associated vasculitis, reinforcing the concept that the subclinical reactivation of CMV is crucial in driving expansion of this T cell subset [205]. A randomized controlled proof-of-concept trial aimed at studying the immunomodulating effect of Valaciclovir in vasculitis is under way [206].

Another potential approach is based on the modulation of CD8+ T-cell exhaustion. This process can develop during chronic viral infection, due to persistent antigen stimulation and inadequate costimulation, and it prevents viral clearance (e.g., HCV) or adequate response to vaccination (e.g., Influenza). This process can be induced by the virus itself, such as in the case of EBV, as a tool to evade immune system [83]. On the other hand, T cell exhaustion is associated with better outcome in several AIDs [207]. 

Interestingly, kidney-infiltrating T cells in LN have a hypofunctional profile caused by an “exhausted” transcriptional signature, possibly representing a tissue defense mechanism to suppress T cell response [208].

A selective induction of this T cell phenotype might represent a new therapeutic tool to blunt autoimmunity, although its potentially detrimental effect on an associated chronic viral infection requires further study. Of interest is the fact that an anti-CD3+ therapy (teplizumab) to promote islet-specific CD8+ T-cell exhaustion showed a beneficial role in type 1 diabetes [209].

### 4.5. Inhibition of TLRs and Other Innate Immunity Modulators

Production of IFN-α and other cytokines by activated pDCs, a key process in SLE pathogenesis, can be determined by EBV-binding to class II MHC molecules, followed by endosomal uptake and TLR-9 (EBV-DNA) or TLR-7 engagement (EBV-encoded RNA, EBER) [69,70,71,144].

NA-sensing TLRs may therefore represent an interesting therapeutic target and oligonucleotide-based TLR antagonists have already proved capable of inhibiting EBV-DNA/RNA induced IFN-α production by human pDCs [145].

Another potential target could be Cathelicidin LL37, an antimicrobial and immunomodulatory peptide which can form complexes with the DNA and autoantibodies (LL37-DNA/anti-DNA). These are endocytosed by pDCs and recognized by TLR-9, leading to DC activation [210].

### 4.6. Inhibition of miRNAs 

An interesting therapeutic target is represented by miR-30e. This is induced by HBV and other viral infections and targets multiple negative regulators of innate immune signaling, thus activating immune response. The inhibition of miR-30e in mouse models and also in SLE patients (for example with anti-HBV treatment) has significantly reduced type I IFN response [211].

### 4.7. Analysis and Modulation of Microbiota

Important research efforts focusing on how human commensal microbes trigger systemic autoimmunity have been made over the last decade [212]. Gut microbial composition and function are markedly altered in untreated SLE patients compared with HCs, having an autoimmunogenic and proinflammatory profile characterized by a consistently reduced Firmicutes-to-Bacteroidetes ratio [213,214].

Gut bacteria can either promote autoimmunity themselves through molecular mimicry, sharing protein epitopes with the Ro60 autoantigen or native DNA, or modulate the risk of autoimmunity-associated viral infections. For example, *Staphylococcus Aureus* colonization reduced the time to CMV acquisition in a prospective birth cohort followed up for 10 years [215]. On the other hand, microbiota has its own virobiota, in the form of bacteriophages which regulate microbiota composition, providing selective advantages to certain strains [216].

These data suggest that the interplay between infections and the immune system should be studied over time by defining individual exposure to bacterial and viral infections and their mutual interactions, long before the clinical onset of autoimmunity. 

Finally, a recent study showed that PBMC virome of SLE patients has a significantly increased diversity compared to HCs and that viral taxa can reliably discriminate cases from controls. These SLE-associated virome signatures might represent a source of biomarkers to predict disease course and outcome [217].

Taken together, these data seem to provide a rationale for microbiome-modulating approaches as adjuvant therapies of AIDs [218].

### 4.8. Antiviral Vaccines 

Another potential approach could be that of directly targeting replication of viruses involved in autoimmunity through vaccines. Vaccination against EBV or EBV-activated HERV in early childhood, at least in groups at high risk, could in theory help prevent the development of SLE at a later age, as epidemiological evidence suggests that a more advanced age of EBV infection might be associated with increased viral replication and risk of autoimmunity induction [219]. The availability of serological or genetic biomarkers to predict risk of evolution towards full-blown AID would be essential for this strategy. However, the vaccine itself could entail risks of autoimmunity if the patient has an intrinsic defect in suppression of postviral inflammation, and not only an abnormal viral replication. This concern has been limiting this approach. Reports of post-HPV vaccination autoimmune phenomena including SLE underlie the need for low-similarity peptide antigens in order to prevent the risk of adverse events triggered by immune cross-reactivity [220].

## 5. Conclusions

Viral infections represent in general a key environmental factor in SLE pathogenesis and can play an important role in triggering disease onset and flareups or in modulating clinical phenotype. Some viruses such as EBV, B19V and HERV can contribute to triggering SLE onset, whereas others, such as CMV, probably modulate disease course, having a more controversial role. Over the last decade, growing evidence has clarified a broad spectrum of viral immune-modulating mechanisms, which deeply alter the host immune system and interact with SLE-intrinsic alterations in B and T cell-mediated responses, as in the case of EBV-driven autoreactive B cell “immortalization”. Classical mechanisms of viral-triggered autoimmunity, such as molecular mimicry, have been further confirmed and a multitude of peptide overlap between viral and self-antigens has widened the concept of SLE as “an autoimmune mosaic syndrome”. Research on the epigenetic modulation of IFN-responsive genes by reactivated HERVs has shed light on this missing link between environmental stimuli and the genome. The relationship between microbiota, virobiota and SLE susceptibility genes is another area of ongoing research. New biomarkers and innovative therapeutic approaches may derive from this progress, including manipulation of innate immunity response, T-lymphocyte subpopulations, and epigenetic mechanisms of viral control on SLE susceptibility genes.

Even if their impact on autoimmunity has been recognized for a long time, viruses are acquiring an increasingly important role as new players in SLE. A better understanding of the interactions between viral proteins and NAs and the host immune system could help improve the management of this heterogeneous and challenging disease.

## Figures and Tables

**Figure 1 viruses-13-00277-f001:**
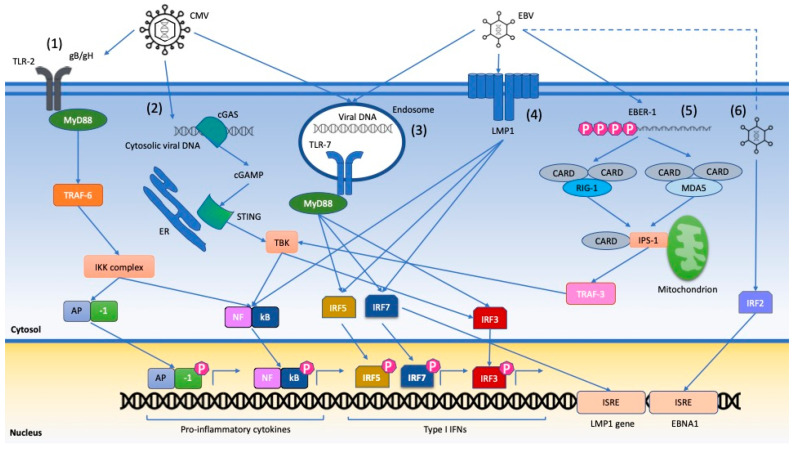
Multiple intracellular pathways triggered by Cytomegalovirus (CMV) and Epstein–Barr virus (EBV) infection converge on expression of proinflammatory cytokines and type I Interferon (IFN) response genes. (1) CMV glycoproteins B and H (gB/gH) engage surface TLR-2 and trigger an inflammatory pathway leading to AP-1 activation; (2) cytosolic CMV-DNA interacts with cyclic GMP-AMP synthase (cGAS) and initiates a cascade that targets NFkB and IRF3; (3) CMV nuclear material trafficked to the endosome triggers TLR-7 signaling, leading to IRF-3, -5 and -7 activation; (4) in EBV latency phase, LMP1 induces expression of IR-5, -7 and NFkB; (5) cytosolic EBER-1 is recognised by RIG-1 and MDA-5 and associates with IPS-1 via CARD-like domains, leading to NFkB and IRF3 activation; (6) EBV induces IRF2 and activates ISRE, which codes for EBNA proteins. Legend: MyD88: Myeloid differentiation primary response 88, IKK complex: IkB kinase complex, TRAF: TNF receptor associated factor, AP-1: activator protein-1, EBER-1: EBV-encoded RNA-1; ER: endoplasmic reticulum, cGAS: cyclic GMP-AMP synthase, cGAMP: cyclic guanosine monophosphate-adenosine monophosphate, STING: cyclic GMP-AMP receptor stimulator of interferon genes, TBK: TANK-binding kinase, IRF: interferon regulator factor, LMP1: Epstein–Barr virus latent membrane protein 1, RIG-1: retinoic acid-inducible gene 1, CARD: caspase activation and recruitment domain, MDA5: melanoma differentiation-associated protein-5, IPS-1: interferon-beta promoter stimulator 1, ISRE: IFN-stimulated response element.

**Figure 2 viruses-13-00277-f002:**
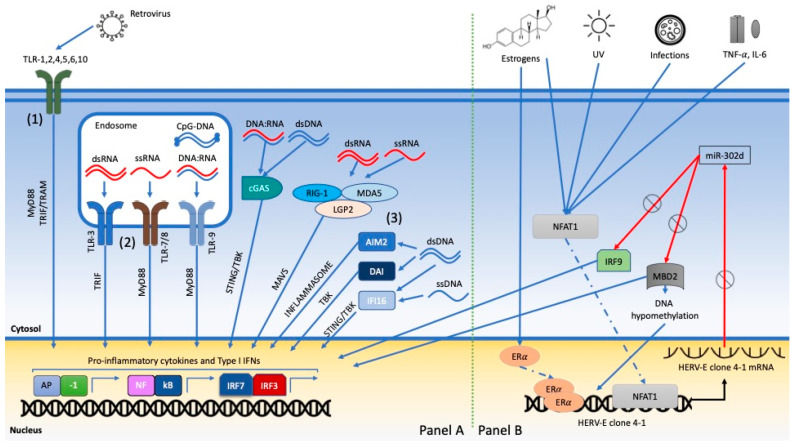
Human Exogenous and Endogenous RVs initiate proinflammatory intracellular pathways converging on critical immune genes involved in SLE pathogenesis. Exogenous RVs RNA and retrotranscribed DNA can activate different cellular sensors located at plasmatic (1) and endosomal membranes (2) or in the cytosol (3), triggering a proinflammatory response (Panel A). HERVs similarly activate endosomal TLRs and also mediate the effects of environmental factors on critical immune genes: estrogens, UVB and infections can increase HERV-E clone 4-1 mRNA expression in SLE CD4+ T-cells and PBMCs, activating a cascade which leads to DNA hypomethylation and type I IFN response (Panel B). Legend: MyD88: Myeloid differentiation primary response 88, TRIF: TIR-domain-containing adapter-inducing interferon-β, TRAM: TRIF-related adaptor molecule, AP-1: activator protein-1, cGAS: cyclic GMP-AMP synthase, cGAMP: cyclic guanosine monophosphate-adenosine monophosphate, STING: cyclic GMP-AMP receptor stimulator of interferon genes, TBK: TANK-binding kinase, IRF: interferon regulator factor, RIG-1: retinoic acid-inducible gene 1, MDA5: melanoma differentiation-associated protein-5, AIM2: absent in melanoma 2, DAI: DNA-dependent activator of interferon regulatory factor, IFI16: γ-interferon-inducible protein IFI-16, MAVS: mitochondrial antiviral signaling protein, LGP2: the laboratory of genetics and physiology 2, NFAT1: nuclear factor of activated T cells 1, MBD2: methyl-CpG-binding domain protein 2,UVB: ultraviolet beams, ERα: estrogen receptor alpha.

**Table 1 viruses-13-00277-t001:** General mechanisms of virus-induced autoimmunity.

Mechanism	Description	References
Molecular mimicry	Viral antigens with structural similarity with self-antigens can be presented to and activate autoreactive T-lymphocytes.	[8,9,14,15,16,17,18]
Epitope-spreading	Over time, persistent viral infection elicits autoantibodies directed not only towards initial antigens but also multiple epitopes of the same antigens or even different antigens, increasing breadth of immune response.	[8,9,14,19]
Superantigen production	Superantigens lack antigenic specificity and bind to T-cell receptor (TCR) and major histocompatibility complex (MHC) class II molecules, activating T-lymphocytes with a wide range of specificities.	[8,9,14]
Bystander activation	Release of cytokine by antigen-presenting cells (APCs) or virus-specific T- lymphocytes activates neighboring preprimed autoreactive T-lymphocytes.	[8,9,14,20]
Altered apoptosis and clearance deficit	Viral infections can increase cell apoptosis, with activation of T-helper 17 (Th17) and release of nondigested nuclear material; if a clearance deficit is present, this process may stimulate autoreactive B-lymphocytes survival.	[8,9,14,22]
Epigenetic factors	DNA methylation, histone modifications and RNA-based mechanisms are the three main epigenetic modalities which allow viruses to modulate expression of genes involved in immune response.	[8,9,14,23,24,25,26]
Persistent or recurrent viral infection	Persistent infection by lymphotropic viruses can stimulate expansion of polyclonal lymphocytes, leading to autoantibody production Recurrent infections can trigger “facilitating antibodies” which enhance inflammation and antigen exposure in subsequent infective episodes, leading to autoimmunity (e.g.,T1D)	[8,9,14,27,28,29,30]
Innate immunity activation	Viral DNA/RNA bind to different PRRs which initiate pathways leading to type I IFN response	[31,32,33]
Direct cytotoxicity	Viruses can infect and directly kill target cells, causing AIDs	[29,34]

**Table 2 viruses-13-00277-t002:** General features of the viruses putatively involved in systemic lupus erythematosus (SLE) pathogenesis.

Virus		Family (Type of Genome)	Proposed Pathogenetic Mechanism in SLE	Therapeutic Approaches	References
Epstein–Barr virus (EBV, HHV4)		Herpesviridae (dsDNA, linear)	Molecular mimicry, epitope spreading	Synthetic nucleoside analogs (acyclovir/ganciclovir) effective against lytic infection only, not recommended. Corticosteroids possibly beneficial in patients with airway defects or EBV-induced autoimmune complications.	[36,37,38]
Cytomegalovirus (CMV, HHV5)		Herpesviridae (dsDNA, linear)	Epitope spreading	Synthetic nucleosides (ganciclovir or valganciclovir) drugs of choice for serious infections or treatment of immunocompromised hosts. Nucleotide (cidofovir) or pyrophosphate analogs (foscarnet) second-choice drugs.	[39,40]
Parvovirus B19 (B19V)		Parvoviridae (positive or negative ssDNA, linear)	Molecular mimicry	Supportive care with transfusion (severe anemia, chronic hemolytic disorders). Reduction of immunosuppression.	[41,42]
Torque Teno Virus (TTV)		Anelloviridae (negative ssDNA, circular)	Molecular mimicry	Not sensitive to current antiviral prophylaxis/therapy. Viral load reduction observed in HIV pts undergoing Highly active antiretroviral therapy (HAART). IFN associated with viral clearance during treatment of coinfecting hepatitis viruses	[43,44]
Hepatitis C virus (HCV)		Flaviviridae (positive ssRNA)	Epitope spreading	Direct-acting antivirals against viral proteases or polymerases. Ribavirin and IFN as second options.	[45]
Dengue virus (DENV)		Flaviviridae (positive ssRNA)	Epitope spreading	No specific treatment	[46,47]
Retroviruses(RVs)	Exogenous retroviruses (HIV, HTLV)	Retroviridae (positive diploid ssRNA, linear)	Dysregulation of apoptosis and molecular mimicry (HIV), regulation of CD4 expression (HTLV-1)	Highly active antiretroviral therapy (HAART) (HIV); nucleoside/nucleotide reverse transcriptase inhibitors associated with IFN useful in HTLV-associated haematological diseases, even if prone to relapses. Insufficient evidence to support use of antiretroviral therapy for the treatment of HTLV-1-associated myelopathy/tropical spastic paraparesis (HAM/TSP)	[48,49,50,51]
	Human Endogenous Retroviruses (HERV)	Retroviridae (integrated in the host genome)	Molecular mimicry, defects in IFN-stimulatory DNA pathways	HAART therapy for HIV may also blunt activation of some HERVs	[43,52,53,54]

**Table 3 viruses-13-00277-t003:** EBV proteins involved in immune system evasion and autoimmunity induction.

Protein	Mechanism	References
Viral IL-10 homologue	It inhibits IFN-γ production, CD8+ cytotoxic T-cells and MHC-I expression	[62]
EA/R	It is a viral Bcl-2 homologue which confers resistance to apoptosis	[64]
LMP-1 and LMP-2A	They rescue infected B cells from apoptosis and are involved in molecular mimicry. LMP-1 triggers IFN-α production by pDCs through TLRs and mediates PDL-1 overexpression in neutrophils. LMP-1 renal expression is increased in LN and correlates with severity.	[17,66,67,68,69,70,71,72,73]
EBNA-1	It can induce autoimmunity through molecular mimicry with C1q	[74,75]
EBNA-2	It is a TF which controls all other latent viral genes. 50% of SLE-predisposing loci can be occupied by EBNA-2, suggesting a key role in AIDs	[76]

**Table 4 viruses-13-00277-t004:** Peculiar alterations of EBV-specific immune response in SLE.

Type of Alteration	Comment	References
Reduced EBV-specific CD8+ lymphocyte response and increased CD4+ lymphocyte response	Impaired cytotoxic potential of EBV-specific CD8+ lymphocytes is due to a SLE-intrinsic defect and is probably the primum movens of altered immune response	[78,79,80,81,82,83]
Decreased Th17 and Treg response	Imbalance between Th17 and T reg is a major cause of AIDs and decrease in Th17 may be an important feature of EBV/CMV infection.	[84]
Elevated EBV viral load in B cells and PBMC and aberrant expression of viral mRNAs of lytic (BZLF-1, BLLF-1) and latent phase proteins (LMP-1, LMP-2 and EBNA-1)	It is due to impaired control of EBV infection by CTLs, with frequent viral reactivations (BZLF-1, BLLF-1) and abnormal latency state (LMP-1, LMP-2, EBNA-1)	[82,85,86,87,88,89]
Elevated EBV seroprevalence	EBV seroprevalence is very high especially in certain populations, in which prior EBV infection appears necessary to develop SLE	[90,91,92,93,94]
Elevated titers of antibody against early lytic (EA/D; EA/R) and latent (EBNA) EBV antigens	This could represent an enhanced compensatory humoral response secondary to an inadequate T-cell control of a chronic EBV infection, with frequent reactivations. It can be associated with production of autoantibodies.	[75,89,91,95,96,97,98]
Elevated prevalence of IgA against EA/D antigen and coexistence of different EBV-specific immunoglobulin isotypes	It may indicate disseminated EBV infection, with higher lytic rate of epithelial cells (IgA) and lymphocytes (IgG).	[89,99,100]
Temporal relationship between anti-EBV antibodies and SLE manifestations	Anti-EBV humoral response can precede SLE onset or flareups. It identifies first-degree unaffected family members of SLE patients who are at risk of transitioning to SLE	[75,101,102,103]
Coexistence of anti-EBV antibodies and SLE-specific autoantibodies	Cross-reactivity between EBV- and self-antigens explains associations between anti-EA/D and anti-Ro/anti-La antibodies, also observed in Sjögren’s syndrome, and between anti-EBNA-1 and anti-C1q.	[74,93,104,105]

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
