# Peer review of "Viral Infections and Systemic Lupus Erythematosus: New Players in an Old Story"

_viruses, 2021, doi:10.3390/v13020277_

Round 1

Reviewer 1 Report

In this review, the authors discuss the association between viral infections and SLE pathogenesis. Though this is not an innovative topic, arguments are exhaustively tackled with hints on future therapeutic perspectives.

General comments:

  1. Being the review specifically focused on SLE, I would have expected the authors to discuss about the pathophysiology, clinical and serologic manifestations and treatments of the disease in the introduction. This information is instead missing and the pathogenetic mechanisms confounding and shifting between SLE and AIDs;
  2. Given the role of viral infections in the pathogenesis of SLE, it would be important to delineate to what extent the immunosuppressive therapies might contribute to this scenario. Vaccinations should be mentioned among the antiviral approaches described in Section 9;
  3. The text reads unduly long and poorly uniform throughout the manuscript. For instance, the part concerning EBV lifecycle sounds too detailed compared with the following sections dealing with other viruses. I would suggest drafting a table reporting the microbiological features, pathogenic implications and marketed or potential therapeutic approaches for the most important viruses described in the text (EBV, PV-B19, RVs, CMV) and removing unessential tables; one or more figures schematically reproducing the pathogenic pathways unleashed by the viruses (e.g., linking viral proteins or nucleic acids to specific downstream cascades) would be useful;
  4. English language should be carefully revised for grammatical and lexical mistakes and typos which may lead to misinterpretation (e.g., the acronym LES incorrectly appears a couple of times; incorrectly spelled terms; missing words, punctuation marks or symbols after CD8 or CD4; convoluted periods; abbreviations, etc.);
  5. Literature search strategy is not described and should be provided.

Specific comments:

  1. Line 2: in the title the abbreviation SLE should be replaced by systemic lupus erythematosus.
  2. The abstract is too long and should be reformulated in order to shortly convey the key messages of the review. The example reported in lines 20,21 should be deleted.
  3. The introduction is too short. SLE pathophysiology, clinical and serologic manifestations and treatments not mentioned. SLE GN should be briefly described as this organ-specific manifestation is often mentioned afterwards. Paragraph 2 could be fused with paragraph 1.
  4. The methods and the aim of this review should be described.
  5. Lines 56-61: The generation of a humoral response is not the only pathway that can be eventually activated following a viral infection. The authors should include other mechanisms, like innate immunity and direct cytotoxicity. Type I IFN production is especially important following the interaction of viral genomes with cell PRRs and should be mentioned in table 1.
  6. Table 2 needs to be removed as the same information can be retrieved from the article by Pan et al., ref. 8.
  7. I would suggest shortening the section 3.1.1 in order to focus on the key message (interaction between EBV and the immune system). Omitted information could be provided in a table or a figure (e.g., mechanisms governing lytic and latent phases and relevant repercussion on the immune system).
  8. Line 87. It should be stated here that EBV can infect T lymphocytes and neutrophils as well (doi: 10.1016/j.jaut.2018.10.025; ref. 39).
  9. Lines 185-186: “This reinforces the concept of a link between SLE-intrinsic immunosuppression, enhanced frequency of EBV reactivation and autoimmunity induction” This sentence is misleading. SLE patients don’t have an intrinsic immunosuppression, rather an unbalanced B/T cell-mediated response towards EBV and other viruses.
  10. Line 213-215: Please specify in the comparison the percentage of healthy controls positive for 2 isotypes; Draborg et al.
  11. Line 220 and 227: “being essential for as EBV-DNA polymerase”; “not only long before onset of SLE”, confounding sentences, please reformulate.
  12. Lines 230-231: I guess the authors refer to the cross-reactivity between EBV antigens and the self-antigens Ro-SSA, La-SSB and C1q. This should be specified. Sjögren’s syndrome should be cited as another example of EBV-triggered cross-reactivity towards Ro-SSA and La-SSB (DOI: 10.1097/BOR.0000000000000622).
  13. Table 4 should specify whether the studies cited have been carried out in a preclinical or clinical setting.
  14. Lines 272-274: the transient appearance in the serum of APL antibodies is a quite common phenomenon during infections. The author should discuss this aspect. Besides, APS is a separate disease, although in some cases it can overlap SLE: doi: 10.5152/eurjrheum.2015.0085. Furthermore, LAC positivity should be included among the serologic parameters along with anti-β2GPI and anti-cardiolipin antibodies.
  15. Line 288: the authors only cite the review by Jia et al. as providing sufficient evidence for linking PV-B19 to SLE without discussing the underneath pathogenetic mechanism.
  16. Line 299: the statement affirming that there are 15 families of HERVs is not correct. According to the classification system adopted, HERVs can include up to 200 families (doi: 10.3389/fonc.2013.00234; doi: 10.1128/JVI.79.1.341-352.2005).
  17. Line 306-307: this statement is misleading. HERVs are part of the human genome. The authors should rewrite this period saying that environmental factors can epigenetically reactivate the transcription of HERV-related sequences. Hence, HERVs may represent the missing link between the genetic predisposition to autoimmunity and external triggers.
  18. Line 360: besides TLR7, the authors should include other RNA-binding PRRs, like TLR3, TLR8, RIG-I and MDA-5, and DNA-binding PRRs like TLR9.
  19. Line 364: not only DNA, also RNA can sensitize PRRs in the cytosol and endosomes.
  20. Line 373: the authors should state here that LTRs are often present as solitary sequences, spread in a high number of copies throughout the human genome and likely controlling the transcription of adjacent genes.
  21. Figure 1 seems to graphically reproduce the results from the article by Wang et al. (ref.110) being therefore poorly informative to the purpose of the review. My suggestion is to implement this figure by providing other pathogenetic mechanisms linking HERVs and exogenous retroviruses to SLE pathogenesis.
  22. Lines 425-428: missing reference. The authors should also discuss the immunosuppressive activity played by the ISU domain of the HERV-H env glycoprotein in autoimmunity, doi: 10.1002/art.39867; doi: 10.1016/j.clim.2018.03.007.
  23. Line 445: CD4+ T lymphocyte may include Treg as well. Maybe the authors refer to CD4+ Th1 lymphocytes?
  24. Line 450: ankylosing spondylitis?
  25. Paragraph 5.3 is redundant and could be omitted or used as an introduction; similarly, paragraphs 6.2 and 7.2 should be deleted and relevant information moved to paragraph 6.1 and 7.1, respectively. Table 5 is unnecessary.
  26. Lines 502-506: redundant and confusing period. The cross-reactivity between the HRES-1/p28 (residues 41-55 and 156-170) and TTV peptide ORF2a is not mentioned (Gergely et al.).
  27. Lines 510-515: please rewrite this period.
  28. Lines 539-540: Accelerator of autoimmunity? Autoantibody production?
  29. Line 561: again, the SLE-intrinsic immunosuppression should be replaced by the dysregulation in the B/T cell responses occurring in this disease.
  30. Line 580: what do the authors mean with “vascular” phenotype? Please specify.
  31. Line 584: HCV is not a lymphotropic virus, rather an hepatotropic virus supposed to have an additional extra-hepatic tropism, including that for B lymphocytes. Why would HCV deserve a separate discussion? I would rather enclose section 8.1 into section 8.2.
  32. 587: Sjögren’s syndrome.
  33. Section 9, Future perspectives: it should be clarified that viral infections and autoimmunity share the activation of overlapping pathogenetic pathways, which would justify the use of antiviral therapeutics in SLE.
  34. Lines 626-627: please specify the clinical context in which this approach has shown successful results (oncogenesis?). Line 636: anti-retroviral agents.
  35. Lines 666-667: please note that TLR3 and TLR7 recognize RNA and not DNA strands.

Author Response

General comments

  1. We devoted part of the Introduction to describing general features of SLE. We clarified that general pathogenetic mechanisms of autoimmunity described in Section 2 are shared between SLE or other AIDs. We expanded the concept of persistent and recurrent viral infection role in autoimmunity and added some examples, as requested by Reviewer 2.
  2. We added a sentence and a reference highlighting the role of chronic immunosuppressive therapy in increasing risk of viral infections in SLE; however, we clarified in the Aim of the Review that this work is focused on the role of viral infections in triggering SLE onset and flares and in modulating its course and not primarily on the impact of immunosuppressive therapy on viral infections. A paragraph about anti-viral vaccinations as new potential approaches to treat SLE has been added in Section 9.
  3. The text was shortened and revised. The part concerning EBV remains more represented than sections dealing with other viruses, mainly because available literature on EBV and SLE is much larger than the one concerning other viruses. We drafted a Table (Table 1) reporting microbiological features, pathogenic implications and therapeutic approaches of viruses described in the text and added two figures schematically reproducing the pathogenic pathways unleashed by some viruses (Fig 1: EBV and CMV; Fig 2: HERV).
  4. English language was revised
  5. Literature search strategy was added

Specific comments:

  1. Line 2: in the title the abbreviation SLE was replaced by systemic lupus erythematosus.
  2. The abstract was shortened and reformulated in order to shortly convey the key messages of the review.
  3. The introduction was expanded including description of general SLE features and Lupus nephritis. We would maintain Paragraph 2 to distinguish general mechanisms of virus-induced autoimmunity in SLE from single viruses specific mechanisms described in Paragraph 3.
  4. The methods (Literature search strategy) and the aim of this review were described.
  5. Lines 56-61: We included two further mechanisms of virus-induced autoimmunity apart from generation of a humoral response - innate immunity activation triggering type I IFN response and direct cytotoxicity - and we mentioned them in Table 1. Table 2 was removed as suggested and content was summarised in the text.
  6. We shortened the Section 3.1.1 in order to focus on the key-message (mechanisms of interaction between EBV and the immune system with analysis of main viral proteins involved in this process).
  7. Line 87. We stated here that EBV can infect T lymphocytes and neutrophils as well and quoted ref 39 of previous version (doi: 10.1016/j.jaut.2018.10.025);
  8. Lines 185-186: we corrected the sentence as suggested,
  9. Line 213-215: we specified in the comparison the percentage of healthy controls positive for 2 isotypes (Draborg et al).
  10. Line 220 and 227: we reformulated sentences correcting mistakes.
  11. Lines 230-231: we tried to clarify the concept of cross-reactivity between EBV antigens and self-antigens Ro-SSA, La-SSB and C1q and quoted Sjögren’s syndrome as another example of EBV-triggered cross-reactivity towards Ro-SSA and La-SSB (Maslinska M. et al. DOI: 10.1097/BOR.0000000000000622).
  12. We specified in the text that studies cited in Table 4 were carried out in a clinical setting.
  13. Lines 272-274: we discussed frequent transient appearance of APL antibodies during infections and the concept of primary or secondary APS, adding the suggested Reference (doi: 10.5152/eurjrheum.2015.0085). We included LAC positivity among the serologic parameters, along with anti-β2GPI and anti-cardiolipin antibodies, as suggested.
  14. Line 288: we modified the text discussing the underneath pathogenetic mechanisms of PV-B19 infection, which support plausibility of a role of this virus in addition to epidemiological association confirmed by the metanalysis.
  15. Line 299: we corrected the sentence of the number of HERVs families and added the two suggested references (doi: 10.3389/fonc.2013.00234; doi: 10.1128/JVI.79.1.341-352.2005).
  16. Line 306-307: we rewrote the period as suggested.
  17. Line 360: we rewrote the paragraph; besides TLR7, we included other RNA-binding PRRs, like TLR3, TLR8, RIG-I and MDA-5, and DNA-binding PRRs like TLR9; 3 references were added
  18. Line 364: we expanded paragraph  on PRRs, included other RNA-binding PRRs (please see answer to point 18)
  19. Line 373: we rewrote the period as suggested
  20. We implemented Figure 1 by providing other pathogenetic mechanisms linking HERVs to SLE pathogenesis.
  21. Lines 425-428: we inserted the missing reference (Emmer A, Staege MS. The retrovirus/superantigen hypothesis of multiple sclerosis. Cell Mol Neurobiol 2014) and   discussed the immunosuppressive activity played by the ISU domain of the HERV-H env glycoprotein in autoimmunity, adding related references (doi: 10.1002/art.39867; doi: 10.1016/j.clim.2018.03.007).
  22. Line 445: we changed “CD4+ T lymphocytes” into “CD4+ Th1 lymphocytes”
  23. Line 450: we corrected “ankylosing spondylitis”
  24. Paragraph 5.3 was omitted; similarly, paragraphs 6.2 and 7.2 were deleted and relevant information moved to paragraph 6.1 and 7.1, respectively. Table 5 was deleted.
  25. Lines 502-506: we rewrote the period and included cross-reactivity between the HRES-1/p28 (residues 41-55 and 156-170) and TTV peptide ORF2a (Gergely et al.).
  26. Lines 510-515: we rewrote the period.
  27. Lines 539-540: we specified the role of phosphoprotein 65 as inducer of autoantibody production
  28. Line 561: we replaced the sentence as suggested.
  29. Line 580: we removed the generic term “vascular” phenotype specifying SLE clinical phenotypes which might be associated with CMV
  30. Line 584: we corrected statements on HCV and also expanded paragraph on HCV (as requested by Reviewer 2); we enclosed all RNA-viruses within section 8.1 and separately discussed DNA-virus Papillomavirus in section 8.2, as requested by Reviewer 2; 
  31. We corrected “Sjögren’s syndrome”.
  32. Section 9, Future perspectives: we clarified that viral infections and autoimmunity share the activation of overlapping pathogenetic pathways, which would justify the use of antiviral drugs in SLE, as suggested.
  33. Lines 626-627: we specified the oncologic context in which this approach has shown successful results. Line 636: we corrected “anti-retroviral agents”.
  34. Lines 666-667: we tried to clarify the role of different TLRs activated by EBV nucleic acids.

Reviewer 2 Report

Well written review. It would be great if the authors could add more examples of not only persistent but also chronically reoccurring viral infections.  Especially infections with RNA viruses, which are known to either persist on RNA level and therefore stimulate type I IFN or where it was shown they lead to autoimmunity in specific organ. Also, it would be great if there could be a better separation in discussion of DNA based viruses and RNA based viruses, which do not integrate into human genome, therefore the mechanism of persistence and stimulation of immune system is less understood. As of now putting Dengue virus together with human papilloma virus in one chapter is confusing. I would suggest separating the RNA and DNA viruses.

Author Response

  1. We added two examples of chronically recurrent viral infections as inducers of autoimmuity and inserted them in Paragraph 1(General mechanisms of virus-induced autoimmunity). We described recent evidence on the role of recurrent Coxsackieviruses B (CV-B) infections in type 1 diabetes (T1D) and the possibile role played by Herpetic viruses (HSV-1 and VZV) in two post-infectious autoimmune neurological disorders (autoimmune encephalitis and neuromyelitis optica). The first example is in our view of particular interest because it provides and explanation of the reason why a chronically recurrent viral infection could eventually trigger an autoimmune disorder, describing the mechanism of “antibody-dependend enhancement” and how “facilitating autoantibodies” lead to infection dissemination, amplification of INFα response with autoantigen exposure and ultimately autoimmune attack on pancreas.

  1. We separated discussion of DNA and RNA-based viruses in Paragraph 8, and expanded paragraph devoted to HCV, adding details about mechanisms exploited by non-retrovirus RNA viruses to achieve “within host” persistent infections without integrating into host genome (both in Section 1/Table 1 and in Section 8.1): hijacking cellular factors including miRNA that bind its genome protecting it from degradation and inhibiting nucleic acid detection by innate immunity sensors.

        In the first part of the Review, we would preserve current sequence of virus (EBV-B19V- RVs- TTV-CMV-other viruses) to emphasize the strongest association of the first three agents with SLE and the progressively less definite role played by the others.   

Round 2

Reviewer 1 Report

The authors extensively revised and improved the manuscript according to the comments and suggestions. Some minor edits to the manuscript are still required.

- General comments:

Please recheck that the use of abbreviations is coherent and accurate throughout the text and tables, without repetition or shifts between full words and abbreviations (some examples are IFN, ANA, RVs, APL, RIG-I, MDA-5, RLRs, BAFF, pDCs, IL, HRES-1, T1D). Similarly, “+” symbols after CD4 or CD8 are still missing either in the text or tables. Please verify and edit.

I would also recommend a second round of whole language revision before resubmission, since some periods still appear convoluted and difficult to understand. Some typos and misspellings are indicated in the points below.

-Specific comments:

  1. Abstract, line 35, and line 164: please replace PV-B19 with B19V. Abbreviations should be coherent throughout the manuscript.
  2. Lines 49-53: since these are crucial aspects discussed afterwards, please mention the impaired clearance of nucleic acids (NA) and the hyperproduction of type I IFN amongst the key-aspects of SLE pathogenesis.
  3. Lines 63-66: please mention the biological agent belimumab among the therapeutic strategies for SLE.
  4. Line 114: viruses instead than virus.
  5. Lines 133-137: the reference 29 (Levet et al.) should be moved to the end of the period dealing with T1D, line 133.
  6. Line 141: please replace danger with damage; lines 142-143: please specify that the term PRRs refers to pattern recognition receptors instead than pathogen-associated molecular pattern receptors.
  7. Line 158: what does SNC stand for? Central nervous system?
  8. Table 1, bystander activation row: cytokine.
  9. Line 172: please specify that neutrophils can be infected by EBV as well.
  10. Line 209-210: through TLRs.
  11. Lines 218-219: please delete “expression intensity” and rewrite this sentence.
  12. Line 223: “cannot not presented”. Please reformulate this sentence.
  13. Figure 1: The localisation of TLR2 in endosomes may sound misleading especially if combined with viral DNA. I would suggest illustrating TLR2 only on the plasma membrane. Please check the caption as well (lines 237-238).
  14. Line 276: please replace independent with regardless. Line 293: SLE patients.
  15. Line 315: worldwide.
  16. Line 318: Misleading sentence: please specify that you are talking about B19V proteins.
  17. Line 339: full-blown SLE; line 342: BM, is it bone marrow? Please specify; line 365: small case series; line 386: sensors; line 439: antigens.
  18. Line 373: Quoting Table 2 instead than Table 1 seems more appropriate in this context. Please note that the improper citation of Table 1 seems to recur several times in the manuscript (e.g., in the TTV and CMV sections).
  19. Line 456: dsDNA sensitises only TLR9.
  20. Lines 458-460: this information has already been provided in section 2. Here the authors should simply state that NAs of RVs can sensitise other PRRs in the cytosol besides TLRs in the endosomes.
  21. Line 488: foreign infections; line 496: please remove “point a”; line 506: in a similar way to.
  22. Line 515: Please rewrite this period: “However, HERVs may also play an immunosuppressive role in SLE”. Line 516: negatively correlates with interleukin-6 (IL6) and TLR7 expression. Line 518: in ex-vivo experiments.
  23. Figure 2: the figure depicts both exogenous and endogenous RVs and the title should be edited accordingly. Please clarify also the mechanism illustrated for exogenous RVs in the figure caption.
  24. Line 568: please remove the term “employment” and reformulate the sentence.
  25. Line 599: “probably, also probably”. Please edit the sentence.
  26. Lines 604-607: this period reports the same information of lines 433-436 and should be deleted.
  27. Line 619: Figure 1 should be quoted at the end of this sentence.
  28. Line 625: please replace LES with SLE.
  29. Lines 637-638 and 652-653: incorrect sentence; please rewrite.
  30. Lines 692-696: convoluted period. Similarly, lines 698-700 should be rewritten. Line 728: full stop missing.
  31. Lines 751-753: please use the simple present tense in this sentence.
  32. Lines 756-757: please rewrite: A randomized controlled proof-of-concept trial aiming at studying (…).
  33. Line 789: significantly reduces; Line 818: suggests; line 826: the risk of adverse events triggered by immune cross-reactivity.
  34. Line 835: The phrase “interact with SLE-intrinsic immunodeficiency” is misleading. Please rewrite.
  35. Line 848: interactions do not solely regard viral proteins but viral nucleic acids as well.
  36. Line 904: signaling; line 907: nucleic acid.
